# Data-Driven Models Informed by Spatiotemporal Mobility Patterns for Understanding Infectious Disease Dynamics

**Die Zhang** [1,2]**, Yong Ge** [1,2,]*****, Xilin Wu** [1,2] **, Haiyan Liu** [3]**, Wenbin Zhang** [1,2] **and Shengjie Lai** [4,5]

1    State Key Laboratory of Resources and Environmental Information System, Institute of Geographic Sciences & Natural Resources Research, Chinese Academy of Sciences, Beijing 100101, China; zhangd@lreis.ac.cn (D.Z.); wuxl.18s@igsnrr.ac.cn (X.W.); zhangwb@lreis.ac.cn (W.Z.)
2    University of Chinese Academy of Sciences, Beijing 100049, China
3    Ocean Data Center, Southern Marine Science and Engineering Guangdong Laboratory (Zhuhai), Zhuhai 519082, China; liuhaiyan@sml-zhuhai.cn
4    WorldPop, School of Geography and Environmental Science, University of Southampton, Southampton SO17 1BJ, UK; shengjie.lai@soton.ac.uk
5    Shanghai Institute of Infectious Disease and Biosecurity, Fudan University, Shanghai 200032, China
*    Correspondence: gey@lreis.ac.cn

**Abstract:** Data-driven approaches predict infectious disease dynamics by considering various factors that influence severity and transmission rates. However, these factors may not fully capture the dynamic nature of disease transmission, limiting prediction accuracy and consistency. Our proposed data-driven approach integrates spatiotemporal human mobility patterns from detailed point-of-interest clustering and population flow data. These patterns inform the creation of mobility-informed risk indices, which serve as auxiliary factors in data-driven models for detecting outbreaks and predicting prevalence trends. We evaluated our approach using real-world COVID-19 outbreaks in Beijing and Guangzhou, China. Incorporating the risk indices, our models successfully identified 87% (95% Confidence Interval: 83–90%) of affected subdistricts in Beijing and Guangzhou. These findings highlight the effectiveness of our approach in identifying high-risk areas for targeted disease containment. Our approach was also tested with COVID-19 prevalence data in the United States, which showed that including the risk indices reduced the mean absolute error and improved the R-squared value for predicting weekly case increases at the county level. It demonstrates applicability for spatiotemporal forecasting of widespread diseases, contributing to routine transmission surveillance. By leveraging comprehensive mobility data, we provide valuable insights to optimize control strategies for emerging infectious diseases and facilitate proactive measures against long-standing diseases.

**Keywords:** human mobility; emerging infectious disease; COVID-19; disease containment; surveillance

## 1. Introduction

High-threat infectious hazards are emerging and re-emerging diseases that may have devastating consequences on health and life in multiple countries or worldwide, such as pandemics [1]. For instance, the outbreak of the severe acute respiratory syndrome coronavirus in 2003, H1N1 influenza in 2009, Ebola virus disease in West Africa in 2013–2016, Zika virus disease in 2015, and the novel coronavirus disease in 2019 (COVID-19) evolved to an unprecedented scale and geographic extent, significantly straining the world's healthcare systems [2]. The occurrence and transmission patterns of infectious diseases are changing as a result of accelerated global integration and the impact of climatic, ecological, and social environmental changes [3,4]. The development of robust predictive models to forecast the dynamics of infectious diseases plays a crucial role in containing their transmission and in real-time surveillance. Furthermore, related findings can further inform policies, such as targeted interventions, mitigation strategies, emergency responses, and allocations of health care resources [5].

Traditional epidemic prediction models have used compartment-based models to estimate disease transmission dynamics at the population level. Examples include the Susceptible-Exposed-Infectious-Removed (SEIR) models and their variants, which have been widely employed to predict the characteristics of the epidemic process [6,7]. However, the design of epidemiological models involves numerous assumptions about disease spread dynamics, and their interpretability and usability have been limited by the underlying assumption of the spatiotemporal homogeneity of the spread of a virus [8]. In practice, disease transmission patterns are substantially heterogeneous in space and over time and correlated with various spatiotemporal driving factors, such as demographic [9], environmental [10–12], social [13], and economic [14] factors. Therefore, a data-driven approach that involves statistical analysis and machine learning has emerged as a tool that can model spatiotemporal patterns of infectious diseases. The machine learning approach has been used to assess factors that place people at a higher risk of measles [15,16], and researchers have worked on influenza forecasting for a long time using statistical and machine learning methods, such as the autoregressive integrated moving average model and random forest algorithm [17]. Statistical and machine learning models have mainly attempted to simulate the effects of driving factors (i.e., predictive variables) on the spread dynamics of infectious diseases [18–20]. However, most relevant driving factors have limited ability to directly reflect the process of infectious disease transmission and the fine-grained details of the spatiotemporal dynamics of outbreaks.

To provide adequate knowledge of the physical dynamics of disease spread in space and over time, several studies have investigated proxy variables informed by physics to promote the positive effects of applying predictive variables in understanding transmission. Typically, human interaction in close physical proximity is the primary cause of the transmission of highly contagious diseases [21,22]. Furthermore, internet users' activity data as surrogate indicators or supplemental data for influenza-like illness activity were investigated to predict influenza epidemics in near real time [23]. These data were widely aggregated from Google searches, Google trends, Wikipedia, and social media (e.g., Twitter and Baidu) to forecast influenza [24,25]. During COVID-19, measuring human interaction was an important step in understanding and predicting the disease's spread [26]. Inter- and intra-county proxies for human interactions through Facebook- and cell phone-derived measures of connectivity and human mobility were suggested as input variables in a machine learning model for predicting county-level COVID-19 cases in the conterminous United States [27]. Moreover, proxies of the pandemic's trajectory were measured by projecting the case and effective reproduction numbers, which were added into a machine learning model to produce the final forecasts on COVID-19 [28].

However, to enhance our understanding of the physical progression of diseases, it is important to optimize proxy indicators that describe human interactions with infected individuals across and within regions. Data pertaining to human movement and contact are valuable in quantifying these interactions and the interconnectedness of different locations, enabling the tracking of an epidemic's trajectory [29,30]. By utilizing time-varying inter-regional population flow and detailed point-of-interest (POI) data [31], it becomes possible to evaluate the spatiotemporal risk of infection in relation to transmission events, while considering individual movement patterns and contact intensity, particularly in relation to infection cases. This assessment of spatiotemporal infection risk has the potential to provide valuable supplementary information, contributing to a more comprehensive understanding of pandemic events.

In the study, we developed two mobility-informed risk indices to describe the risk of infectious disease transmission in space and time. These risk indices were combined with other relevant variables and used in statistical regression and machine learning models to understand disease dynamics for transmission containment and real-time surveillance. The proposed method can be used to detect outbreaks caused by newly introduced acute human-to-human transmitted diseases (e.g., COVID-19 and pandemic influenza) at the early stage of the outbreak, and to predict short-term trends of transmission in community

hotspots where populations have not yet acquired herd immunity. We tested the method using real-world data on COVID-19 outbreaks in Chinese cities and the United States. The results showed that the proposed method was effective in identifying high-risk areas throughout an outbreak in a city, assisting in the implementation of interventions to quickly control the disease spread. Furthermore, the method maintained a generally high level of performance for one- to four-week-ahead forecasts of the county-level COVID-19 prevalence in the United States, contributing to real-time surveillance of disease dynamics within the country.

## 2. Materials and Methods

The aim of this study was to comprehend infectious disease dynamics using statistical and machine learning models based on mobility-informed risk indices, as illustrated in Figure 1. Initially, we developed these risk indices by analyzing individuals' movements and contacts over space and time. To predict infectious disease dynamics, we combined these risk indices with socio-economic, demographic, environmental, and epidemiological factors as predictive variables. We further utilized statistical regression and random forest models to establish the relationships between the predictive and target variables of interest. To validate the models, we used real-world COVID-19 transmission data under two scenarios. At the early stages of an outbreak, timely identification of potential infections in space is critical to contain disease spread. Therefore, we employed the models to identify affected subdistricts during real-world importation-related COVID-19 outbreaks in Chinese cities. Moreover, during widespread community transmission, most areas within a country report continuous increases in infection rates. In this case, we made one- and four-week-ahead forecasts of COVID-19 prevalence at the county level in the contiguous United States to support routine surveillance. For each scenario, we used 10-fold cross-validation for model training and tuning by randomly splitting historical sample data into training and test sets. Finally, we applied the best-tuned model to predict actual COVID-19 transmission dynamics, and prediction performance was estimated by comparing the predicted and actual results.

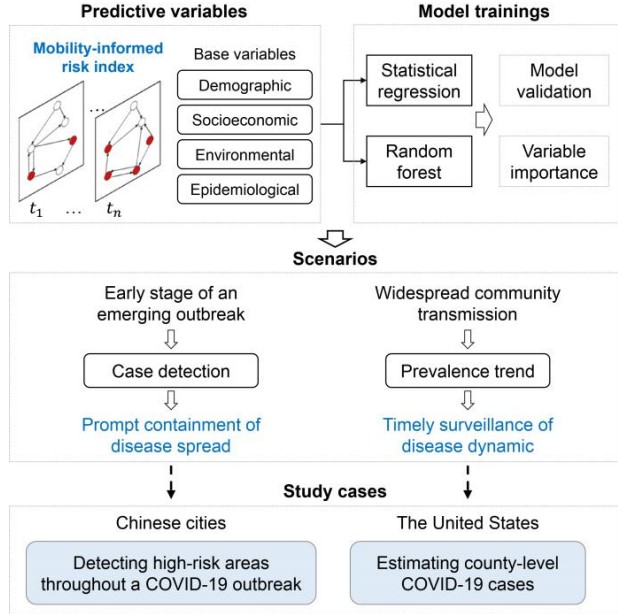

**Figure 1.** Flowchart for estimating the dynamics of human-to-human disease transmission using statistical and random forest models based on mobility-informed risk indices. The models were evaluated under two study scenarios: detection of potential affected subdistricts during COVID-19 outbreaks in Chinese cities; and spatiotemporal forecasts of county-level COVID-19 cases in the contiguous United States.

All data processing, calculation of mobility-informed risk indices, model training and testing for statistical regression, and random forest models—as well as result evaluations—were conducted using the Python programming language. Python provides a robust and widely used platform for data analysis and modeling in the field of statistics.

### 2.1. Mobility-Informed Risk Indices

Transmission risks of infectious diseases were evaluated by examining individual mobility patterns and contact intensity, utilizing data on time-varying population flow, detailed POIs, and the locations of the first confirmed cases (Figure 2). Specifically, the case flow intensity (CFI), which considers the location of initial cases and their movements across regions (such as subdistricts or counties), was derived using an established travel network. The CFI quantifies regional infection risk by counting the cumulative number of initial cases that visited a region; higher CFI values indicate regions that have been visited by more initial cases. Based on the CFI, the case transmission intensity (CTI) was computed to represent the risk introduced by both inter-regional movements and intra-regional contact with initial cases. The CTI is based on the number of potential new infections resulting from the activity of initial cases, and regions with a larger CTI are more likely to have a higher number of infected individuals.

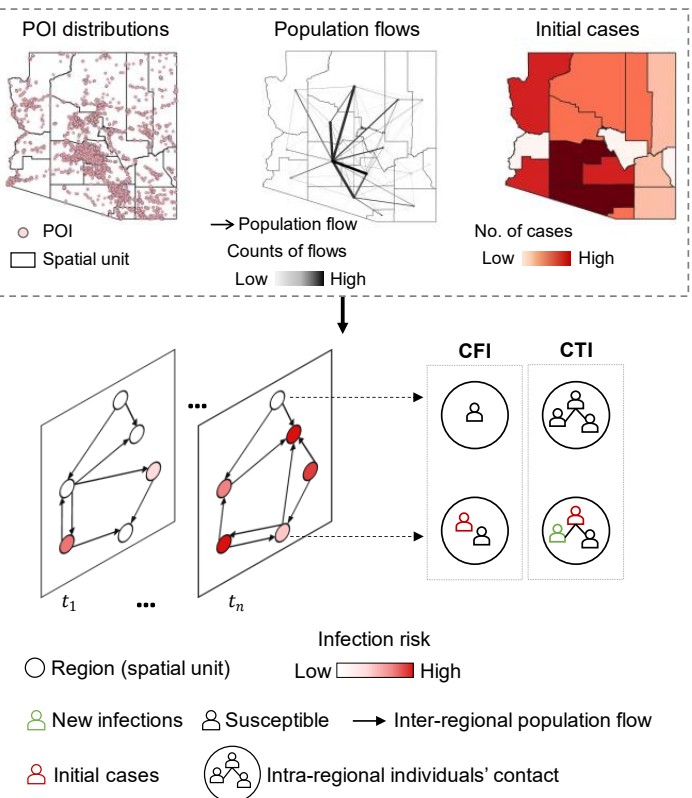

**Figure 2.** Illustration of mobility-informed risk indices. Based on point of interest (POI) data, mobile travel flows, and the locations of initial confirmed cases, two spatiotemporal risk indices were designed: case flow intensity (CFI) and case transmission intensity (CTI).

We expressed all regions as the set $R = \{r_i, i = 1, 2, \ldots, N\}$. At hour $t$, regions from which people go to region $r_i$ are denoted as $F_{\rightarrow i}^t = \{r_j \in R, r_j \neq r_i, 0 \leq |F_{\rightarrow i}^t| < N\}$, where $|F_{\rightarrow i}^t|$ is the number of elements in the set. Regions where people travel from region $s_i$ are denoted as $F_{i \rightarrow}^t = \{r_k \in R, r_k \neq r_i, 0 \leq |F_{i \rightarrow}^t| < N\}$. The number of visitors from $s_j$ to $s_i$ is $P_{ji}^t$. Accordingly, the population size in region $s_i$ at hour $t$ can be computed by:

$$P_i^t = P_i^{t-1} + P_{\rightarrow i}^t - P_{i \rightarrow}^t, \tag{1}$$

where $P_{\to i}^t = \sum_{r_j \in F_{\to i}^t} P_{ji}^t$ and $P_{i\to}^t = \sum_{r_k \in F_{i\to}^t} P_{ik}^t$ is the population that moves in and out of a region $s_i$, respectively.

The CFI-based regional infection risk can be depicted by the hourly cumulative counts of initial cases. At hour $t-1$, there were $C_j^{t-1}$ and $C_i^{t-1}$ initial cases in regions $s_j$ and $s_i$, respectively. At hour $t$, $P_{ji}^t$ visitors went to region $s_i$ from $s_j$, of which the number of initial cases was positively proportional to the population flow given by:

$$C_{ji}^t = C_j^{t-1} \cdot \frac{P_{ji}^t}{P_j^{t-1}}, \tag{2}$$

The hourly number of the initial cases is expressed as:

$$C_i^t = C_i^{t-1} + \sum_{r_j \in F_{\to i}^t} C_{ji}^t - \sum_{r_k \in F_{i\to}^t} C_{ik}^t, \tag{3}$$

where $\sum_{r_j \in F_{\to i}^t} C_{ji}^t$ and $\sum_{r_k \in F_{i\to}^t} C_{ik}^t$ is the total number of initial cases entered and left region $s_i$ at hour $t$.

Therefore, the CFI risk index can be computed by:

$$\mu_i^D(\text{CFI}) = \sum_{t=0}^{D} C_i^t, \tag{4}$$

where $D$ is the duration of the population flow under consideration.

Instead, the infection risk based on CTI was depicted by hourly cumulative counts of potential new infections due to contact with the initial cases within a region. At hour $t$, the number of new infections increases in region $s_i$ in terms of intra-regional contact with $C_i^t$ initial cases. The infection rate is given by:

$$\lambda_i = \beta_i \cdot \frac{C_i^t}{P_i^t}, \tag{5}$$

where $\beta_i$ is the intra-regional transmission rate derived from the logged POI-based diversity index. That is, $\beta_i = \left( \sum_c (m_{i,c})^q \right)^{1/(1-q)}$, where $m_{i,c}$ is the number of POIs in region $s_i$ for POI secondary category $c$, and $q$ is the exponential factor equal to 0.4 [32]. POI-based diversity indices have been widely used to depict neighborhood vibrancy and human activity [33–35].

Therefore, the CTI risk index can be computed by:

$$\mu_i^D(\text{CTI}) = \sum_{t=0}^{D} I_i^t, \tag{6}$$

where $I_i^t$ is the number of new infections in region $s_i$ at hour $t$ and is expressed as $I_i^t \sim Binom\left(P_i^t - C_i^t, \lambda_i\right)$ [26].

## 2.2. Models for Predicting High-Risk Subdistricts in Chinese Cities

This section presents a method for assessing outbreaks of newly emerging or emergent acute human-to-human transmitted diseases in a city. We propose using logistic regression and random forest classifiers based on CFI and CTI risk indices at the initial outbreak stage to predict which subdistricts are at risk of being affected throughout an outbreak. To outline our method, we collected data on actual COVID-19 outbreaks in Beijing and Guangzhou and used an epidemiological model to simulate various outbreak scenarios at the subdistrict level in these cities as the sample data. Using the sample for training and tuning, we computed CFI and CTI risk indices which were inputted into logistic regression and random forest classifiers to determine which subdistricts were affected. Finally, we applied the fitted models to predict the affected subdistricts in actual COVID-19 outbreaks

in Beijing and Guangzhou, and we evaluated the accuracy of our predictions by comparing them to real-world data.

### 2.2.1. Data on COVID-19 Outbreaks

At the initial stage of the COVID-19 outbreak, prompt interventions were implemented to control the spread, and as such, socioeconomic and environmental factors had little impact on the occurrence of transmission events. We obtained data on mobility, points of interest (POI), demographics, and epidemiological outbreaks in Beijing and Guangzhou. Given the challenges in accessing detailed information on infected individuals and fine-grained origin-destination human mobility, we relied on metapopulation-based data at the township-level divisions (i.e., subdistricts) of the two cities as our unit of analysis. Beijing consisted of 331 subdistricts, while Guangzhou had 168 subdistricts. Information on affected subdistricts and the number of cases were obtained from press releases and daily pandemic notification reports released by the Beijing and Guangzhou Municipal Health Commissions (Table A1). From 11 June 2020 to 5 July 2020, a total of 368 cases across 52 affected subdistricts were reported in Beijing. On 21 May 2021, the index case of a highly transmissible variant of SARS-CoV-2 (VOC Delta) was confirmed in Guangzhou, and by 18 June 2021, 16 subdistricts had been affected with a total of 152 cases (including confirmed and asymptomatic infections). Population data for 2021 at a 100-meter resolution were obtained from WorldPop (www.worldpop.org, accessed on 1 June 2022).

We analyzed anonymized population movement flows between subdistricts in Beijing and Guangzhou using hourly data aggregated from cellular signaling data provided by China Mobile (www.chinamobileltd.com, accessed on 1 June 2022), one of China's largest national mobile carriers. Specifically, we used hourly two-day data from 11–12 June 2020 to capture people's movement between subdistricts in Beijing before the implementation of travel restrictions across cities due to COVID-19 outbreaks. For Guangzhou, we used hourly inter-subdistrict population flow data from 21–22 May 2021. Additionally, we obtained POI data for 2020 from AMap Services (ditu.amap.com, accessed on 15 June 2022), one of China's main location-based service providers. AMap divided the POI into 23 primary categories, 241 secondary categories, and 2035 tertiary categories.

### 2.2.2. Sample Data Simulated by SEIR Model

As access to sophisticated historical real-world population movement and epidemiological data was limited, we employed a travel network-based SEIR modeling framework [36] to simulate COVID-19 transmission across various outbreak scenarios in Beijing and Guangzhou. The simulated epidemiological data was then used to develop predictive models for subdistricts that would be affected by an outbreak. The SEIR model (github.com/wpgp/BEARmod, accessed on 10 July 2020) is capable of simulating COVID-19 propagation across subdistricts within a city (Appendix B). For a single simulation, the start date was the day when the first confirmed case was infected, and the start location within the subdistricts was chosen randomly. The epidemiological parameters in the model were defined based on existing studies (Table A4). The SEIR model estimated the daily number of new cases in each subdistrict to determine the number of affected subdistricts throughout an outbreak. To generate a series of epidemiological data, we utilized various simulations for each city under different levels of transmissibility and random source locations. Table 1 shows that COVID-19 transmission was simulated at 30 random initial outbreak locations for each of the three different transmission levels controlled by the basic regeneration number ($R_0$), generating a total of 90 sets of COVID-19 outbreak epidemic data for each city. The time-series changes in the number of daily cases under simulated epidemics are shown in Figure A1.

The primary predictive variables used in our study were population, population density, number of POIs, POI density, population flow volume, and mobility-informed risk indices. We calculated the mobility-informed risk indices, namely CFI and CTI, based on hourly flow data in the first two days after an outbreak occurred, and data on the location

of initial confirmed cases, to determine the initial stage risk levels. As an example, for the Beijing outbreak, we defined the spatial unit $s_i$ as one of the 331 subdistricts, and the time unit $t$ as an hour over the 48-hour period of 11–12 June 2020 (i.e., duration $D = 48$). $P_i^{t=0}$ represents the population count in subdistrict $s_i$, and $C_i^{t=0}$ represents the cumulative number of confirmed cases from 13 to 15 June 2020. Using these values and the definition of CFI and CTI, we calculated the spatiotemporal population $P_i^t$, number of cases $C_i^t$, and number of potential infections $I_i^t$ to obtain the early-stage CFI and CTI risks, $\mu_i^D(\text{CFI})$ and $\mu_i^D(\text{CTI})$. We followed a similar process to calculate the CFI and CTI risk values for the Guangzhou outbreak.

**Table 1.** Simulated COVID-19 outbreaks under different transmissibility levels and source locations in Beijing and Guangzhou. For each city, there were three $R_0$ ($R_0$ mean and 95% confidence interval upper and lower thresholds), and the outbreak was considered to start in one randomly selected subdistrict.

| City | $R_0$ | Number of Affected Subdistricts |
|:---:|:---:|:---:|
| Beijing | 3.32 | 43 (95% CI: 37–49) |
| | 1.4 | 14 (12–17) |
| | 3.9 | 52 (38–67) |
| Guangzhou | 4.9 | 26 (22–29) |
| | 3.1 | 4 (3–5) |
| | 6.5 | 93 (81–104) |

$R_0$: basic reproduction number.

### 2.2.3. Logistic Regression and Random Forest Classifier

Simulated data was utilized to develop and refine logistic regression and random forest classifiers for the prediction of subdistricts affected by actual COVID-19 outbreaks in Beijing and Guangzhou. To train and optimize the models, a 10-fold cross-validation approach was employed, whereby the simulated epidemiological data was randomly divided into training and test sets. This cross-validation technique significantly aids in mitigating the risk of overfitting [37]. The tuned model was subsequently utilized to classify subdistricts within each city as either affected or unaffected. The predictions were then compared against actual data regarding affected subdistricts, and the performance of the prediction model was assessed using a confusion matrix. The confusion matrix, a two-by-two table generated by a binary classifier, presents four possible outcomes [38]. Of particular interest to us were two key metrics: sensitivity (SE) and specificity (SP). Sensitivity was calculated as the ratio of correctly estimated affected subdistricts to the total number of actual affected subdistricts, while specificity was calculated as the ratio of correctly estimated unaffected subdistricts to the total number of actual unaffected subdistricts [39]. In a similar vein, Moulaei et al. [40] employed machine learning algorithms to predict COVID-19 mortality and assessed model performance using metrics derived from the confusion matrix such as accuracy, sensitivity, precision, specificity, and receiver operating characteristic (ROC). Likewise, Jahangiri et al. [41] conducted sensitivity and specificity analyses to investigate the impact of ambient temperature and population size on the COVID-19 transmission rate in various provinces of Iran, employing ROC to assess the performance of their classification model, utilizing the confusion matrix.

Additionally, we evaluated the importance of all variables using the permutation feature importance technique, which is defined as the decrease in the model score when a single feature value was randomly shuffled [42].

### 2.3. Models for Estimating COVID-19 Cases in the United States

This section introduces regression models that utilize mobility-informed risk indices to predict the spread of diseases in areas with high prevalence. The models aim to forecast one- to four-week incidences at the county level in the contiguous United States. To achieve this, we computed CFI and CTI risk indices with one- to four-week temporal lags based on daily population flows across counties and reported weekly confirmed cases. These

indices served as inputs for the elastic net and random forest regression models, which were trained and tuned using the log-transformed incidence rate as the target variable. Moreover, the models incorporated multiple base predictive variables selected from a previous study. The fitted models generated one- to four-week ahead forecasts of weekly increases in the number of cases on a given date, and we evaluated the predictive performance by comparing the estimated results with the actual number of confirmed cases.

### 2.3.1. Data on COVID-19 Prevalence

County-level daily COVID-19 cases in the United States were collected from USA Facts (usafacts.org/visualizations/coronavirus-covid-19-spread-map, accessed on 15 July 2022) for the period between 29 March 2020, and 9 April 2021. USA Facts is a reputable non-profit organization that provides data on government tax revenues, expenditures, and outcomes. The data on COVID-19 cases from USA Facts have been widely used in previous studies on COVID-19 spread characteristics [43,44]. Figure A2 shows the weekly number of new cases, with approximately 200,000 cases reported during the week of 29 March 2020 to 4 April 2020, affecting nearly 70% of counties across the country. The number of new cases continued to rise throughout 2020, with 1.5 million new cases per week by the end of the year. Population mobility and POI data were obtained from SafeGraph, which provided precise global POI data. We obtained the aggregated population flow data between counties, covering 22 January 2020, to 15 April 2021, from the website (gis.cas.sc.edu/GeoAnalytics/od.html#, accessed on 10 July 2022). This data included daily origin-destination (OD) mobility data at the county level. SafeGraph's website (www.SafeGraph.com/products/places, accessed on 10 July 2022) provided the spatial distribution of POIs across the contiguous United States, which included 6,778,576 POIs, covering 199 main categories and 400 subcategories.

### 2.3.2. Target and Predictive Variables

To train and tune our models, we used the natural logarithm of new cases per 10,000 people plus one (to avoid zero values) as the target variable. The rationale behind using the log-transformed target variable, as opposed to directly predicting the number of weekly new cases, was to minimize skewness and, more importantly, reduce the sensitivity of the models to the population of counties [27]. The formulas used to calculate the values are as follows:

$$incidence\ rate_i^T = \frac{Cases_i^T}{P_i}, \tag{7}$$

$$y_i^T = \ln\left(incidence\ rate_i^T + 1\right), \tag{8}$$

where $Cases_i^T$ denotes the number of weekly new confirmed cases (from day $T$ to $T + 7$) for the start day $T$, and $y_i^T$ is the log-transformed incidence rate as the target variable for model training. For a given date $T$, the corresponding target variables for one- to four-week-ahead forecasts are the incidence rate by week, with the time range from $T$ to $T + 7$, $T + 7$ to $T + 14$, $T + 14$ to $T + 21$, and $T + 21$ to $T + 28$, respectively.

Studies have shown that the time between exposure to the virus and symptom onset can be up to 14 days [45]. We used a 14-day period of population movement to calculate the CFI and CTI risk indices, with the number of cases reported in the week prior to the forecast date as initial cases. Specifically, we defined $s_i$ as the spatial unit representing a county in the contiguous United States, and $t$ as the time unit representing a given day within the 14-day period of population flows ($D = 14$). To calculate county-level risk values with a one-week temporal lag (CFI_T_1 and CTI_T_1), we set $t = 0$ as the day $T - 14$ for forecast date $T$, and used the weekly number of confirmed cases reported, given by $C_i^{t=0} = Cases_i^{T-7}$, as the initial cases. $P_i^{t=0}$ represented the population size of county $s_i$. We obtained CFI_T_1 and CTI_T_1 based on the definition of CFI and CTI using daily population flows across counties. We divided these values by the county population and recorded them as IN_CFI_T_1 and IN_CTI_T_1. To obtain the corresponding risk values with a two- to four-week temporal lag, we advanced the time of population flows and

initial cases by one week at a time. For example, to calculate CFI_T_1 for a given date $T$, we used the daily inter-county population flow data from $T - 14$ to $T$ and the cumulative number of cases from $T - 7$ to $T$, as shown in Table 2. However, to calculate the risk values with a four-week temporal lag (CFI_T_4 and CTI_T_4), we used the daily inter-county population flow data between $T - 35$ and $T - 21$ and the cumulative number of cases from $T - 28$ to $T - 21$.

**Table 2.** The time range of population flow and initial case data required for calculating the mobility risk index in one- to four-week temporal lags.

| Mobility-Informed Risk Index | Temporal Lag | Duration for Mobility Data | Duration for Case Data |
|---|---|---|---|
| CFI_T_1 | One-week | $(T - 14) \sim (T)$ | $(T - 7) \sim (T)$ |
| CFI_T_2 | Two-week | $(T - 21) \sim (T - 7)$ | $(T - 14) \sim (T - 7)$ |
| CFI_T_3 | Three-week | $(T - 28) \sim (T - 14)$ | $(T - 21) \sim (T - 14)$ |
| CFI_T_4 | Four-week | $(T - 35) \sim (T - 21)$ | $(T - 28) \sim (T - 21)$ |

CFI_T_1, CFI_T_2, CFI_T_3, and CFI_T_4: CFI risk indices with one- to four-week temporal lags.

Basic predictive variables were extracted from two previous studies that were similar to our study. These variables were employed to conduct reference experiments, designated as $REF_1$ and $REF_2$, respectively. $REF_1$ [27] employed various demographic and socioeconomic variables, temperature data, features obtained from Facebook and SafeGraph, and weekly changes in cumulative COVID-19 cases as predictive variables (Table A2). $REF_2$ [28] included a comprehensive set of features such as population health, demographic data, COVID-19 testing results, and projections of the number of cases and Rt (Table A3). Our proposed models incorporated the variables used in each reference study with temporally lagged weekly CFI and CTI risk indices. We named this combination of variables the proposed experiments, which we labeled as $Proposed_1$ and $Proposed_2$. The performance of the proposed models was compared with that of the reference experiments under the same settings, except for the predictive variables utilized (Table 3). While $REF_1$ and $Proposed_1$ were used to forecast 39 consecutive weekly intervals from 3 May 2020 to 24 January 2021, $REF_2$ and $Proposed_2$ were utilized to predict 11 consecutive weekly intervals from 1 November 2020 to 10 January 2021.

**Table 3.** Design for comparison between proposed models and reference experiments.

| Experiment | Predictive Variables Used | Forecast Date | Model |
|---|---|---|---|
| $REF_1$ | See Table A2 | 39 weekly intervals from 3 May 2020 to 24 January 2021 | |
| $Proposed_1$ | Variables in $REF_1$ and mobility-informed risk indices | | Elastic net and random forest regression |
| $REF_2$ | See Table A3 | 11 weekly intervals from 1 November 2020 to 10 January 2021 | |
| $Proposed_2$ | Variables in $REF_2$ and mobility-informed risk indices | | |

### 2.3.3. Elastic Net and Random Forest Regression

We developed a spatiotemporally autoregressive model that can forecast weekly increases in COVID-19 cases up to four weeks ahead, covering 3103 counties. The study involved 39 forecast dates in $REF_1$ and 11 forecast dates in $REF_2$. To train and fine-tune the model, we collected two weeks of historical data preceding each forecast date, resulting in $3103 \times 2$ total samples. The model's performance was evaluated in predicting new cases for the upcoming week using 10-fold cross-validation. For example, to make a one-week-ahead forecast on 3 May 2020, sample data was collected from 19–25 April 2020 and 26 April–2 May 2020, for training and fine-tuning. The fine-tuned model was then used to predict the new cases in each county during the 3–9 May 2020 period. Similarly, to make a

four-weeks-ahead forecast on 3 May 2020, we collected sample data from 29 March–4 April 2020 and 5–11 April 2020, and predicted the number of increased cases between 24–30 May 2020. This training and testing process was conducted for different prediction horizons, and calculated target variables separately for each prediction horizon.

We selected elastic net regression [46] and random forest regression models which addressed the multicollinearity issue among predictive variables. The trained models made forecasts for county-level weekly increases in cases on each forecast date, ranging from one to four weeks in advance. The accuracy of the proposed model's forecasts was evaluated against actual case counts by calculating mean absolute error (MAE) and R-square ($R^2$) values. Additionally, we calculated the permutation importance of all variables used in elastic net and random forest regression models.

## 3. Results

### 3.1. Risk Deification at the Initial Outbreak Stage

Based on the proposed mobility-informed indices, the logistic regression and random forest classifiers demonstrated the ability to identify a range of 50–90% of affected subdistricts during COVID-19 outbreaks in Beijing and Guangzhou, as presented in Table 4. These models exhibited a high accuracy rate in Beijing, correctly detecting over 87% of subdistricts with cases (sensitivity). Additionally, both models demonstrated accurate identification of unaffected subdistricts in Beijing, with a specificity exceeding 0.75. In comparison to the SEIR epidemiological model, the proposed models outperformed in predicting affected subdistricts during the Beijing outbreak. The SEIR model failed to identify 24 out of 52 affected subdistricts, resulting in a sensitivity of only 54%. When considering the trade-off between sensitivity and specificity, the proposed models achieved an Area Under the Curve (AUC) value exceeding 0.9 in the Receiver Operating Characteristic (ROC) analysis for Beijing (refer to Figure 3). Regarding the outbreak in Guangzhou, the logistic regression model demonstrated superior performance in identifying affected subdistricts compared to the random forest classifier, as evidenced by a larger AUC value. It successfully captured 87% of the affected subdistricts. In contrast, the SEIR model exhibited a higher specificity of 0.92 compared to the logistic regression model.

**Table 4.** Performance evaluation of the proposed models in identifying affected subdistricts during real-world COVID-19 outbreaks in Beijing and Guangzhou. The logistic regression and random forest classifier, based on mobility-informed indices, were used to predict subdistricts with COVID-19 cases. The predicted results were evaluated using the confusion matrix. The brackets refer to the 95% confidence interval.

| Model | Subdistrict | Actual COVID-19 Outbreak | | | |
| --- | --- | --- | --- | --- | --- |
| | | Beijing | | Guangzhou | |
| | Estimated / Reported | Affected | Unaffected | Affected | Unaffected |
| Logistic regression | Affected | 45 | 53 | 14 | 58 |
| | Unaffected | 7 | 226 | 2 | 94 |
| | | SE: 0.87 (0.83–0.90) | | SE: 0.87 (0.84–0.90) | |
| | | SP: 0.81 (0.80–0.81) | | SP: 0.62 (0.61–0.62) | |
| | | Affected | Unaffected | Affected | Unaffected |
| Random forest classifier | Affected | 47 | 71 | 8 | 50 |
| | Unaffected | 5 | 208 | 8 | 102 |
| | | SE: 0.90 (0.88–0.93) | | SE: 0.50 (0.47–0.53) | |
| | | SP: 0.75 (0.74–0.75) | | SP: 0.67 (0.66–0.68) | |
| | | Affected | Unaffected | Affected | Unaffected |
| SEIR model | Affected | 28 | 13 | 13 | 12 |
| | Unaffected | 24 | 266 | 3 | 140 |
| | | SE: 0.54 (0.50–0.56) | | SE: 0.81 (0.79–0.83) | |
| | | SP: 0.95 (0.93–0.96) | | SP: 0.92 (0.90–0.94) | |

SE: sensitivity; SP: specificity.

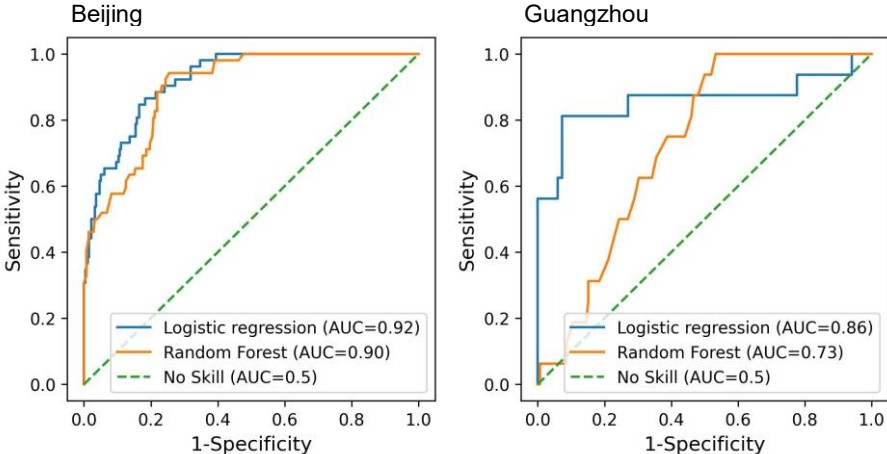

**Figure 3.** Receiver Operating Characteristic (ROC) curves and the corresponding Area Under the Curve (AUC) values for the logistic regression and random forest classifier models. These models utilize mobility-informed indices to predict subdistricts with COVID-19 cases. The ROC curves illustrate the performance of the models in terms of sensitivity and specificity, while the AUC values provide a quantitative measure of their predictive accuracy.

The relative importance of the predictive variables showed that the CFI risk index had the most dominant impact in estimating potential subdistricts with cases throughout an outbreak (Figure 4). Population size and CTI risk index were also significant variables, especially in predicting outbreaks in Beijing.

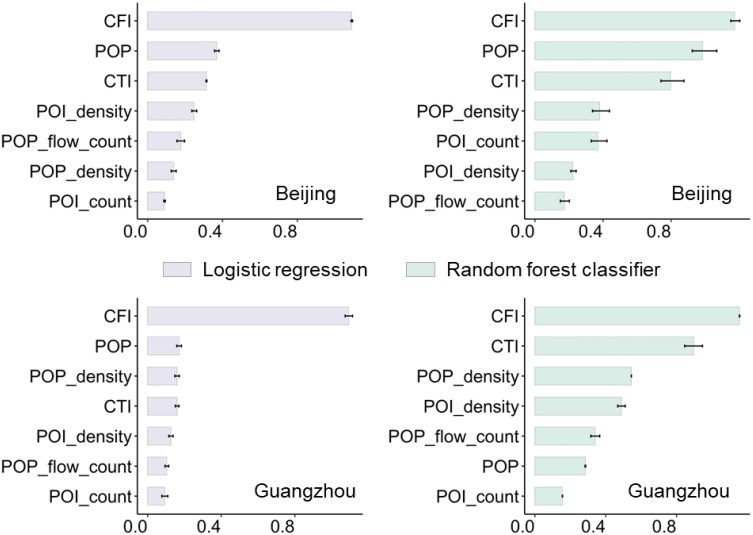

**Figure 4.** Relative permutation importance of predictive variables. The logistic regression and random forest classifier were applied to predict subdistricts with cases of actual COVID-19 outbreaks in Beijing and Guangzhou, respectively. Error bars represent 95% confidence intervals.

### 3.2. Forecasts of Weekly Increased Cases

#### 3.2.1. Forecasting Performance

The proposed models, which incorporate CFI and CTI risk indices, led to a decrease in MAE for most forecast dates compared to the reference studies (Figure 5). Specifically, the proposed method using predictive variables in $REF_1$ ($Proposed_1$) had a lower MAE than $REF_1$ for one-week-ahead forecasts on 28 (95% CI: 2–36) and 37 (33–37) of 39 dates using the elastic net and random forest models, respectively (Figure 5a). On average, $REF_1$ had a higher MAE for four-weeks-ahead forecasts on 20 and 24 dates using the two models (Figure 5d). For one- to four-weeks-ahead forecasts, using the random forest regression,

the average MAE decrease from $REF_1$ to $Proposed_1$ on a given date was 7.5 (3.6–11.4), 2.7 (−3.5–8.9), 1.1 (−5.8–8.1), and 0.4 (−7.8–8.6) (Figure 5a–d). Furthermore, the incorporation of a regression forecasting model, in combination with CFI and CTI related risk indices, led to a decrease in MAE of $REF_2$ for forecasts ranging from one to four weeks in advance. Specifically, the utilization of elastic net regression resulted in MAE reductions for 9, 9, 7, and 7 out of the 11 dates, respectively (Figure 4e–h). Similarly, employing random forest regression in conjunction with CFI and CTI related risk indices resulted in MAE reductions of 9, 7, 7, and 7 out of the 11 dates for forecasts spanning one to four weeks ahead, respectively.

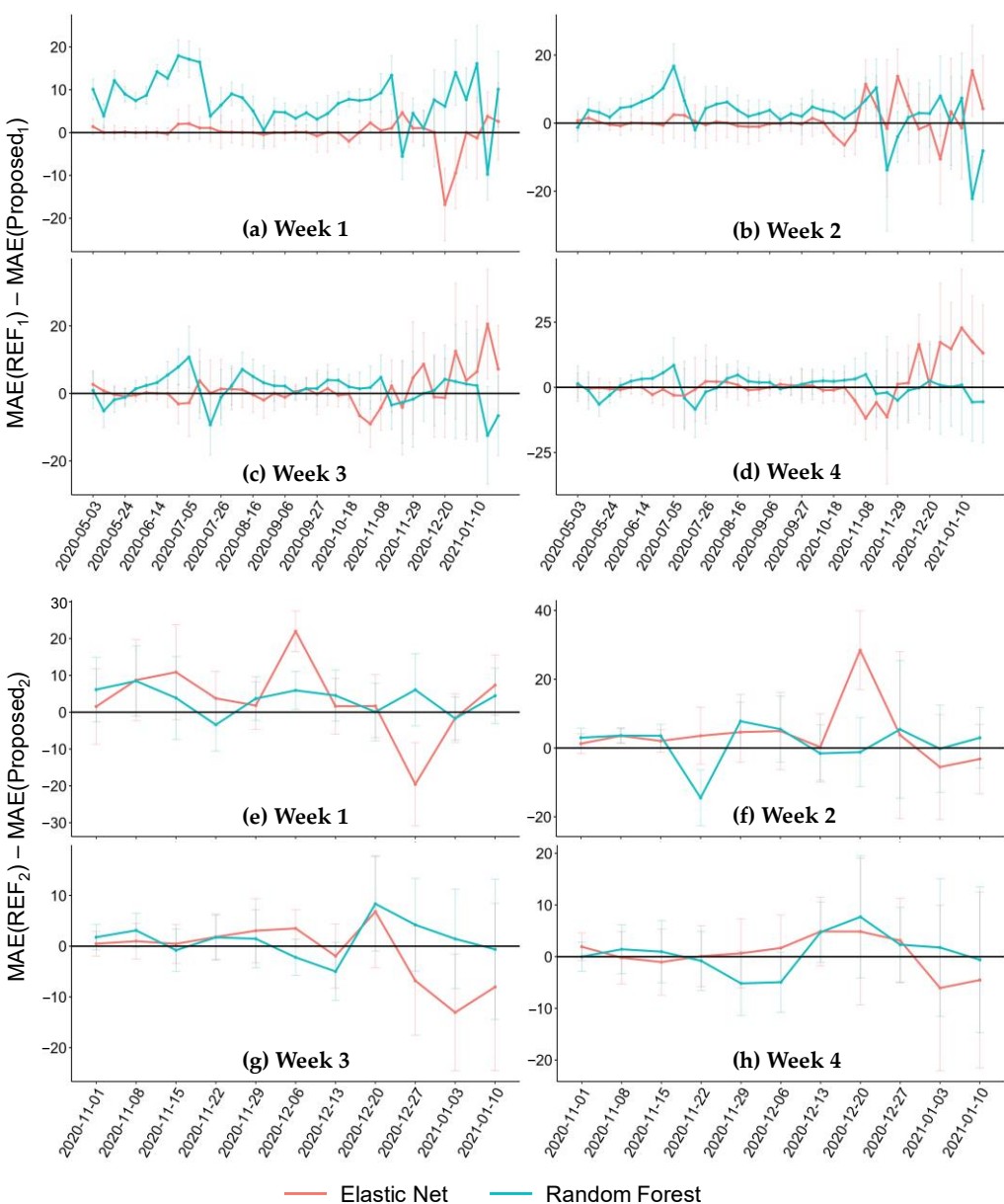

**Figure 5.** Evaluation of models' forecasting performance using the mean absolute error (MAE). For each forecast date, the elastic net and random forest regression were used to predict the weekly increases in the number of cases in U.S. counties for one to four weeks ahead. The MAE was used to measure the error between the estimated and actual number of cases. The proposed method with additional mobility-informed risk indices ($Proposed_1$ and $Proposed_2$) was compared to two reference studies ($REF_1$ and $REF_2$), respectively: (**a**–**d**) the MAE difference between $Proposed_1$ and $REF_1$ for 39 weekly forecast dates and (**e**–**h**) the MAE difference between $Proposed_2$ and $REF_2$ for 11 weekly forecast dates. The error bars represent 95% confidence intervals.

The inclusion of CFI and CTI related variables as inputs in the elastic net and random forest models demonstrated improved $R^2$ for most forecast dates. The Proposed$_1$ method achieved an $R^2$ higher than 0.5 for one- to four-weeks-ahead forecasts on 39 dates, as shown in Figure A3. Similarly, the Proposed$_2$ method, which used random forest regression, achieved an $R^2$ higher than 0.8 for the forecasts on 11 dates. On average, the Proposed$_1$ method improved $R^2$ for one- to four-weeks-ahead forecasts on 35, 31, 31, and 31 dates compared to REF$_1$ when using random forest regression (Figure 6). Additionally, Proposed$_2$ improved $R^2$ on 8, 9, 8, and 7 dates against REF$_2$.

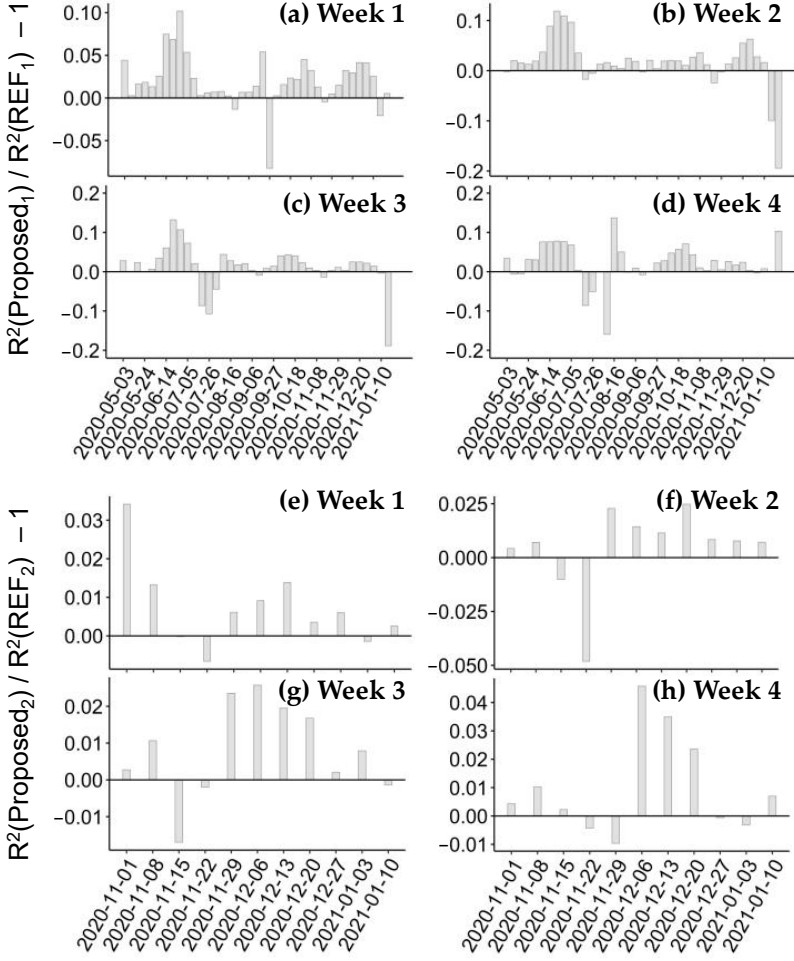

**Figure 6.** Evaluation of models' forecasting performance using the R-square ($R^2$). For each forecast date, the elastic net and random forest regression were used to predict the weekly increases in the number of cases in U.S. counties for one to four weeks ahead. The average value of $R^2$ between the estimated and actual number of cases was calculated. The proposed method with additional mobility-informed risk indices (Proposed$_1$ and Proposed$_2$) was compared to two reference studies (REF$_1$ and REF$_2$), respectively: (**a**–**d**) the $R^2$ difference between Proposed$_1$ and REF$_1$ for 39 weekly forecast dates and (**e**–**h**) the $R^2$ difference between Proposed$_2$ and REF$_2$ for 11 weekly forecast dates.

### 3.2.2. Applicability Analysis

The use of random forest regression on the same forecast dates revealed that while the $R^2$ of REF$_2$ was higher, its MAE was generally greater when compared to REF$_1$ (Figure 7). The average increase in MAE from REF$_1$ to REF$_2$ was −1.6, 23.8, 37.8, and 33.6 for one- to four-weeks-ahead forecasts, respectively, while $R^2$ increased by 0.04, 0.04, 0.03, and 0.07. The incorporation of CFI and CTI risk indices into REF$_1$ had a more pronounced effect on reducing MAE and increasing $R^2$ than incorporating them into REF$_2$. For instance, the forecasts from REF$_1$ to Proposed$_1$ exhibited a greater decrease in MAE and increase in $R^2$

compared to those from $REF_2$ to $Proposed_2$. Additionally, the changes in MAE and $R^2$ over time from $REF_1$ to $Proposed_1$ were smoother for three- and four-weeks-ahead forecasts.

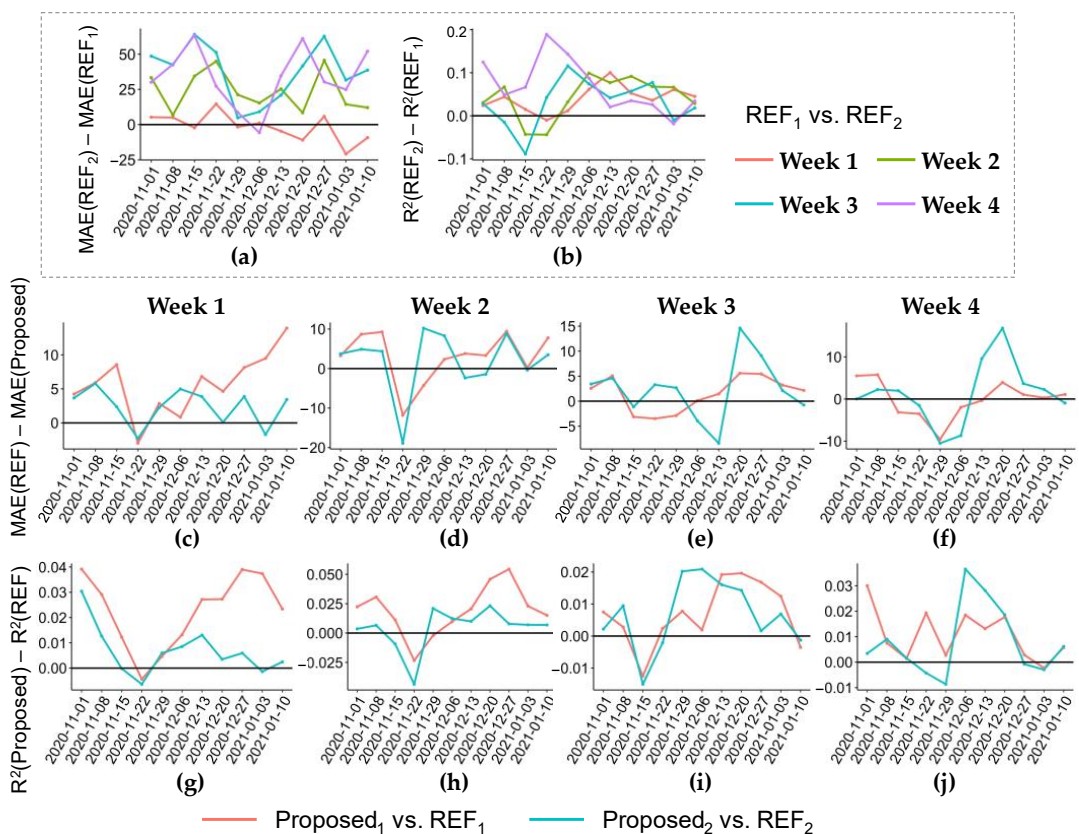

**Figure 7.** Comparison of the forecasting performance between two reference studies using mean absolute error (MAE) and R-square ($R^2$). The weekly increases in the number of cases in U.S. counties were predicted for one to four weeks ahead, and MAE and $R^2$ between the estimated and actual number of cases were calculated. The proposed method was compared to two reference studies, $REF_1$ and $REF_2$, respectively. There were 11 same weekly forecast dates from 1 November 2020 to 10 January 2021: (**a,b**) the MAE and $R^2$ difference between $REF_1$ and $REF_2$ for one- to four-weeks-ahead forecasts using random forest regression; (**c–f**) and (**g–j**), The MAE and $R^2$ difference between the proposed method and reference study using random forest regression, respectively.

The top eight important variables for one- to four-weeks-ahead forecasts consistently included several CFI and CTI related risk variables in the proposed methods (Figures 8 and 9). The variable with the highest importance ranking in the $Proposed_1$ method was the incidence rate in the week before the forecast date (i.e., LOG_DELTA_INC_RATE_T_1). However, the method also gave high importance to CFI and CTI risk indices, such as IN_CFI_T and IN_CTI_T. Moreover, the $Proposed_2$ method consistently gave high importance to state-level test numbers, test positivity, and the predictions of case and Rt alignment, particularly when using elastic net regression. The random forest regression prioritized a few CFI and CTI related variables for one- and two-weeks-ahead forecasts.

The findings indicate that the inclusion of CFI and CTI risk indices can enhance the precision of COVID-19 forecasting models. The models that included CFI and CTI related variables with elastic net and random forest regression methods demonstrated a reduced MAE and an increased $R^2$ in comparison to the reference experiments. The impact of incorporating CFI and CTI was more notable in diminishing the MAE and elevating the $R^2$ of $REF_1$ than $REF_2$. The significance ranking of variables revealed that CFI and CTI indices were among the top predictors of COVID-19 prevalence, particularly in the proposed method that utilized predictive variables in $REF_1$.

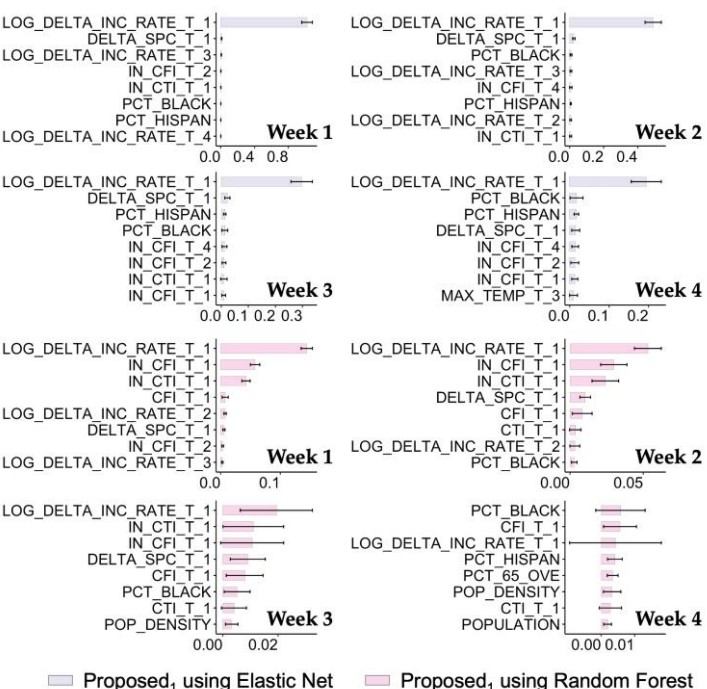

**Figure 8.** Relative permutation importance of the top eight predictive variables for the proposed method. The elastic net and random forest regression were used to predict the weekly increases in the number of cases in U.S. counties for one to four weeks ahead. Proposed$_1$ represents the proposed method based on mobility-informed risk indices and the variables used in a reference (REF$_1$). Error bars represent 95% confidence intervals.

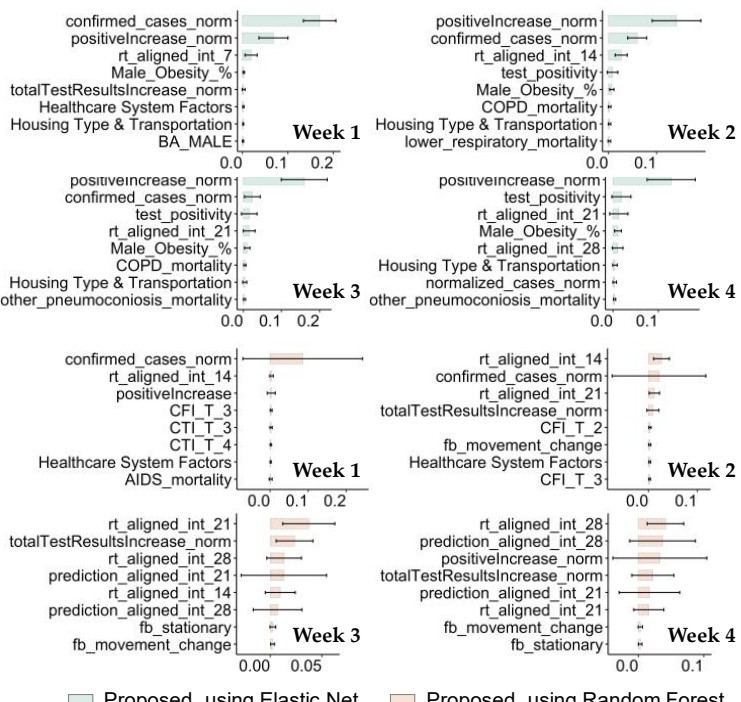

**Figure 9.** Relative permutation importance of the top eight predictive variables for the proposed method. The elastic net and random forest regression were used to predict the weekly increases in the number of cases in U.S. counties for one to four weeks ahead. Proposed$_2$ represents the proposed method based on mobility-informed risk indices and the variables used in a reference (REF$_2$). Error bars represent 95% confidence intervals.

## 4. Discussion

Diverse multidimensional factors may contribute to the severity and rate of disease spread [47,48]. Examining and developing variables that account for the physics of the disease spread process can improve the effectiveness and physical consistency of applying predictive variables to understand the dynamics of infectious diseases. Based on human mobility and POI information data, which are widely used to understand the diffusion of infectious diseases [49,50], our study created mobility-informed risk indices (CFI and CTI) by integrating inter-regional movement and the locations of infections. We further revealed that CFI and CTI indices could effectively identify high-risk areas to help contain COVID-19 spread at the early stages of an emerging outbreak, as well as maintain a high accuracy rate for one- to four-weeks-ahead forecasts of disease transmission.

The timely spatial prediction of infections in the early stages of an emerging outbreak using our proposed method can provide valuable insights for the implementation of interventions aimed at containing disease spread. For instance, interventions such as testing, resource allocation, travel restrictions, and school and workplace closures can be optimized and targeted in 87% of the actual affected subdistricts, as predicted by the logistic regression model based on mobility-informed risk indices for real-world outbreaks in Beijing and Guangzhou (see Table 4). In contrast to traditional SEIR models, which heavily rely on various epidemiological assumptions and parameters that may not be easily or quickly confirmed in the early stage of a pandemic, our method is based on mobility-informed risk variables and has fewer epidemiological assumptions and parameters. This makes it more consistent and easier to use in various cases, especially when rapid response decision-making is required to determine where interventions should be prioritized. Furthermore, our approach accounts for the complex geographic drivers of spatiotemporal heterogeneity, thereby providing accurate predictions of disease transmission.

The incorporation of CFI and CTI risk indices, which account for the physics of disease spread, can significantly enhance the spatiotemporal prediction of the prevalence of infectious diseases. Data analysis on COVID-19 prevalence in the United States show that when the physics of disease dynamics involved in the predictive variables used were less accounted for, the addition of CFI and CTI could greatly improve forecasting performance (as seen for $REF_1$ in Figure 7), indicating that these risk indices provide valuable supplemental physical information. While $REF_2$ used the projections of the case and the effective reproduction number that involved much physical information and had $R^2$ greater than 0.8, the addition of mobility-informed indices still slightly improved forecasting performance. However, the improvement was reduced as the forecasting horizon extended from one to four weeks (as seen in Figure 5). In practice, CFI and CTI mainly reflected the spatial risk in the two weeks following the forecast date and showed more obvious improvement for one- and two-weeks-ahead forecasts than for three- and four-weeks-ahead forecasts. Nonetheless, when other variables covered limited physics-related information, CFI and CTI related variables showed higher relative importance for three- and four-weeks-ahead forecasts (as seen in Figures 8 and 9). In summary, physics-informed factors, such as mobility-informed risk indices, are essential in ensuring the accuracy of disease prevalence predictions. The incorporation of CFI and CTI risk variables can improve forecasting performance, particularly when other predictive variables have limited physics-related information.

This study has several limitations that should be noted. First, while the CFI and CTI variables were able to capture potential intra-regional infectious risks through the establishment of the population flow network, they did not fully account for the risk of infection events that may occur when an infected person moves between two regions. Future research could optimize these indices by considering more risk events. Second, the proposed method requires a training set generated by SEIR model simulations for the identification of high-risk areas at the early stage of an outbreak. The prediction accuracy could be further improved with available historical real-world epidemiological data that are often used in decision-making in the early stage of an outbreak. Third, the effectiveness of

our approach has been successfully validated by analyzing two scenarios of the COVID-19 pandemic in China and the United States. However, to further examine the real-world effectiveness of these approaches, it would be beneficial to obtain more data on various infectious diseases. By obtaining appropriate data support from other countries and regions, our method can be extended to comprehensively understand the dynamics of infectious diseases in diverse contexts beyond those studied. Finally, despite the use of variable selection methods such as the elastic net and random forest regression, multicollinearity among extensive variables may have influenced the permutation importance ranking. Thus, variable filtering could be conducted before model training for case forecasts in the United States to mitigate the effects of multicollinearity.

## 5. Conclusions

The development of robust and efficient predictive models to forecast the dynamics of infectious diseases is crucial for timely and targeted interventions in mitigating and monitoring the impact of disease outbreaks and epidemics. A data-driven approach provides rapid predictions, enabling a timely comprehension of the dynamics of both emerging and persistent infections. By utilizing mobility-informed risk indices, an accurate portrayal of the risk associated with spatiotemporal propagation events is achieved. These indices furnish a priori information pertaining to the physical aspects of disease transmission, thereby enhancing the prediction accuracy and physical consistency of data-driven models. While SEIR models have found extensive application in comprehending infectious diseases, this study also underscores the potential of machine learning and statistical regression models in disease control and surveillance, particularly in complex and multidimensional data scenarios. In conclusion, a data-driven approach, informed by priori physical information, holds promise in contributing to the detection and response of infectious disease outbreaks and epidemics.

**Author Contributions:** Conceptualization, Die Zhang, Yong Ge, and Shengjie Lai; methodology, Die Zhang, Yong Ge, Wenbin Zhang, and Shengjie Lai; software, Die Zhang; validation, Die Zhang, Yong Ge, and Shengjie Lai; formal analysis, Die Zhang; investigation, Die Zhang and Xilin Wu; resources, Yong Ge, Shengjie Lai, and Haiyan Liu; writing—original draft preparation, Die Zhang; writing—review and editing, Yong Ge, Wenbin Zhang, and Shengjie Lai; visualization, Die Zhang and Xilin Wu; supervision, Yong Ge and Shengjie Lai; project administration, Die Zhang and Yong Ge; funding acquisition, Yong Ge and Shengjie Lai. All authors have read and agreed to the published version of the manuscript.

**Funding:** Shengjie Lai is supported by funding from the National Institutes of Health (grant number R01AI160780), the Bill & Melinda Gates Foundation (grant number INV-024911) and the European Union Horizon 2020 (grant number MOOD 874850). Yong Ge was supported by the National Natural Science Foundation of China (grant number 41725006 and 42230110). The funders of the study had no role in study design, data collection, data analysis, data interpretation, or writing of the report. The corresponding authors had full access to all the data in the study and had final responsibility for the decision to submit for publication. The views expressed in this article are those of the authors and do not represent any official policy. The APC was funded by the National Natural Science Foundation for Distinguished Young Scholars of China (grant number 41725006).

**Data Availability Statement:** The main text provides detailed explanations regarding various types of data sources and their download links, with the majority being publicly accessible. Nevertheless, it should be noted that the data concerning anonymized population movement flows and points of interest (POIs) in Beijing and Guangzhou, China are not publicly accessible due to rigorous licensing agreements. However, the aggregated and processed data utilized and analyzed in the present study can be obtained from the corresponding author upon a reasonable request.

**Conflicts of Interest:** The authors declare no conflict of interest.

## Appendix A. Extended Figures and Tables

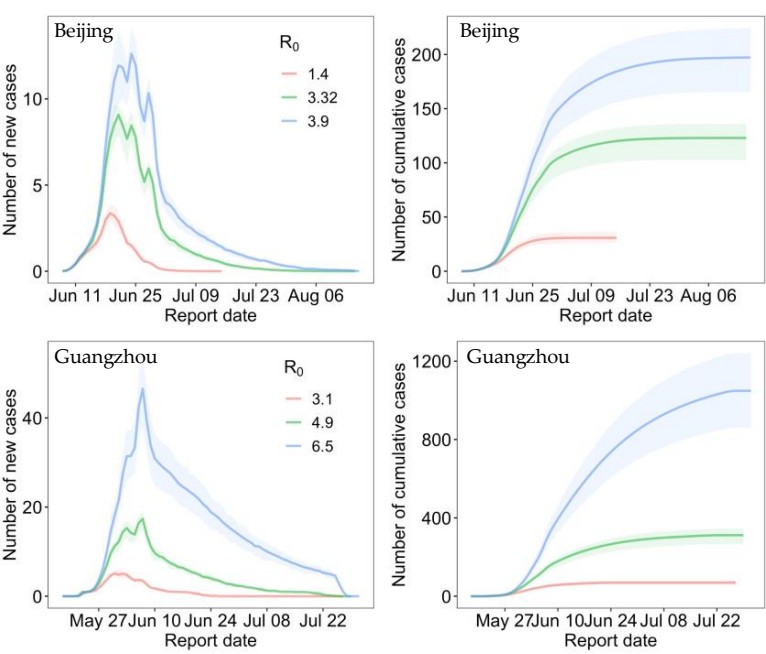

**Figure A1.** Pandemic curves of simulated COVID-19 outbreaks under different values of $R_0$ using the constructed SEIR model in Guangzhou and Beijing. The daily number of new cases and cumulative cases in the two cities are shown, respectively. The simulated transmissions are presented as the mean (solid blue lines) and 95% confidence intervals (shading) of various random source locations of outbreaks.

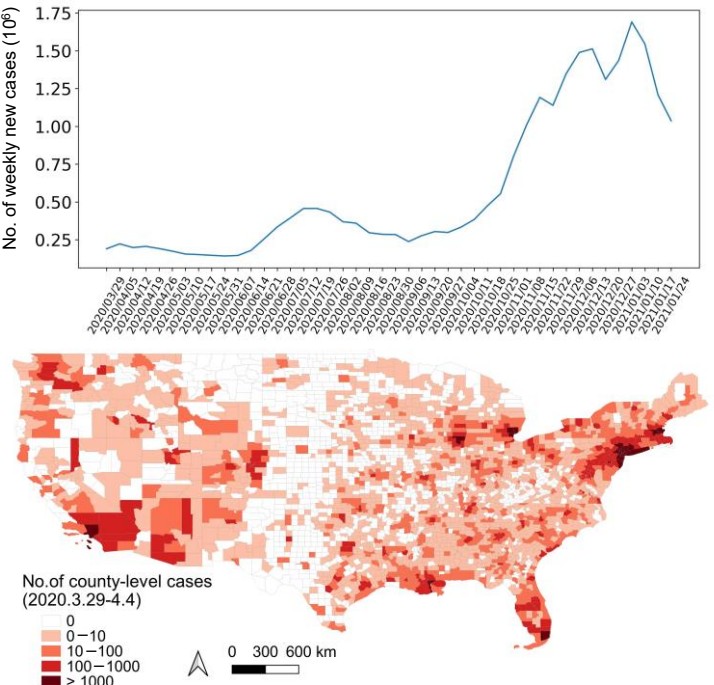

**Figure A2.** A time series of the total weekly number of newly increased COVID-19 cases in the contiguous United States, along with the county-level spatial distribution of the cumulative number of cases from 29 March to 4 April 2020. The data used for this analysis was obtained from USA Facts (usafacts.org/visualizations/coronavirus-covid-19-spread-map, accessed on 15 July 2022), a reliable source of information on COVID-19 cases.

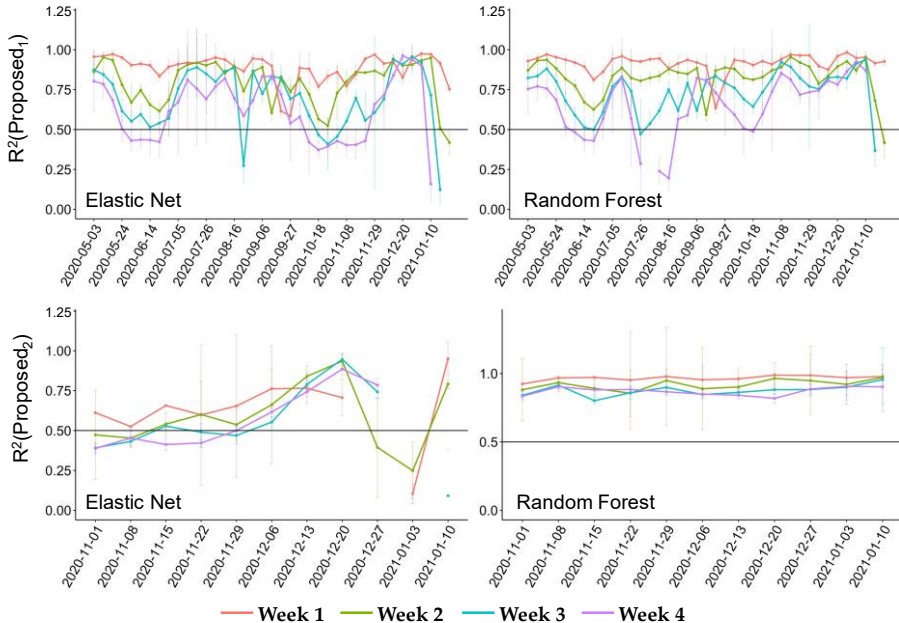

**Figure A3.** R-square($R^2$) of the proposed method on 39 weekly forecast dates in $REF_1$ and on 11 weekly forecast dates in $REF_2$, respectively. Forecasts with $R^2$ less than 0 would not be displayed in the panels.

**Table A1.** Number of cases in each subdistrict affected by COVID-19 outbreaks in Guangzhou and Beijing. There were 152 cases in 16 Guangzhou communities and 368 cases in 52 Beijing communities.

| Subdistrict Name | Number of Cases | Subdistrict Name | Number of Cases |
|---|---|---|---|
| Guangzhou (21 May–18 June 2021) | | | |
| Baihedong Subdistrict | 91 | Longjin Subdistrict | 2 |
| Zhongnan Subdistrict | 29 | Taihe Town | 1 |
| Zhujiang Subdistrict | 10 | Changgang Subdistrict | 1 |
| Ruibao Subdistrict | 4 | Haichuang Subdistrict | 1 |
| Dongjiao Subdistrict | 3 | Nanhuaxi Subdistrict | 1 |
| Dashi Subdistrict | 2 | Beijing Subdistrict | 1 |
| Luopu Subdistrict | 2 | Dongsha Subdistrict | 1 |
| Yongping Subdistrict | 2 | Chongkou Subdistrict | 1 |
| Beijing (11 June–5 July 2020) | | | |
| Huaxiang Area | 192 | Changxindian town | 1 |
| Xihongmen Area | 25 | Changxindian Subdistrict | 1 |
| Xincun Subdistrict | 21 | Yuetan Subdistrict | 1 |
| Huangcun Area | 17 | Youanmen Subdistrict | 1 |
| Yongdinglu Subdistrict | 10 | Yongdingmenwai Subdistrict | 1 |
| Qingyuan Subdistrict | 9 | Yizhuang Area | 1 |
| Lugouqiao Area | 8 | Xingfeng Subdistrict | 1 |
| Majiabao Subdistrict | 7 | Xiaohongmen Area | 1 |
| Tiancunlu Subdistrict | 6 | Wanshoulu Subdistrict | 1 |
| Nanyuan Subdistrict | 6 | Tiantan Subdistrict | 1 |
| Changyang town | 4 | Taipingqiao Subdistrict | 1 |
| Qingyundian town | 4 | Sijiqing Area | 1 |
| Xiluoyuan Subdistrict | 3 | Shibalidian Area | 1 |
| Weishanzhuang town | 3 | Qinglongqiao Subdistrict | 1 |
| Nanyuan Area | 3 | Panggezhuang town | 1 |
| Lugouqiao Subdistrict | 3 | Lixian town | 1 |
| Dahongmen Subdistrict | 3 | Jiugong Area | 1 |
| Zhanlan Road Subdistrict | 2 | Jinrong Street Subdistrict | 1 |
| Yongding Area | 2 | Huilongguan Area | 1 |
| Tiangongyuan Subdistrict | 2 | Hepingli Subdistrict | 1 |
| Linxiao Road Subdistrict | 2 | Guang'anmenwai Subdistrict | 1 |
| Guanyinsi Subdistrict | 2 | Guang'anmennei Subdistrict | 1 |
| Fengtai Subdistrict | 2 | Beizangcun town | 1 |
| Beiyuan Subdistrict | 2 | Balizhuang Subdistrict | 1 |
| Beixinqiao Subdistrict | 2 | Babaoshan Subdistrict | 1 |
| Baizhifang Subdistrict | 2 | Anding town | 1 |

**Table A2.** Predictive variables used in reference study REF$_1$.

| Category | Variable | Abbreviation |
|---|---|---|
| Socioeconomic and demographic | Population density<br>Pct. of African American population<br>Pct. of the male population<br>Pct. of the population aged > 65<br>Pct. of Hispanic population<br>Pct. of the rural population<br>Pct. of Native American population<br>Median household income<br>Pct. of the population with a college degree<br>Pct. of the population who voted republican | POP_DENSITY<br>PCT_BLACK<br>PCT_MALE<br>PCT_65_OVE<br>PCT_HISPAN<br>PCT_RURAL<br>PCT_AMIND<br>MED_HOS_IN<br>PCT_COL_DE<br>PCT_TRUMP_ |
| Temperature | Average of daily minimum temperature in one week<br>Average of daily maximum temperature in one week | MIN_TEMP_T<br>MAX_TEMP_T |
| COVID-19 incidence rate | Natural logarithm of cumulative incidence rate in one week | LOG_DELTA_INC_RATE |
| Features derived from Facebook | Intra-county movement features<br>Inter-county features | RATIO_MOB_T, REL_MOB_T<br>SPC_T |
| Features derived from SafeGraph | Intra-county movement features<br><br>Inter-county features | distance_traveled_from_home,<br>median_home_dwell_time,<br>pct_completely_home_device_count,<br>pct_delivery_behavior_devices,<br>pct_full_time_work_behavior_devices,<br>pct_part_time_work_behavior_devices<br>FPC_T |

**Table A3.** Predictive variables used in reference study REF$_2$.

| Category | Variable | Abbreviation |
|---|---|---|
| Population health | Infectious disease mortality rates (tuberculosis, AIDS, diarrheal disease, lower respiratory disease, meningitis, hepatitis) | AIDS_mortality, diarrheal_mortality, hepatitis_mortality, tubercolosis_mortality, meningitis_mortality, hepatitis_mortality |
| | Respiratory disease mortality rates (interstitial lung disease, asthma, coal pneumoconiosis, asbestosis, silicosis, pneumoconiosis, COPD, chronic respiratory disease, other pneumoconiosis, other respiratory diseases) | COPD_mortality, asbestosis_mortality, asthma_mortality, chronic_respiratory_mortality, coal_pneumoconiosis_mortality, lower_respiratory_mortality, other_resp_mortality, interstitial_lung_mortality, other_pneumoconiosis_mortality, silicosis_mortality, pneumoconiosis_mortality |
| | Mortality risk (0–5, 5–25, 25–45, 45–65, and 65–85 age groups) | mortality_risk |
| | Life expectancy | life_expectancy |
| | Diabetes prevalence rates | Diabetes_Prevalence_Both_Sexes |
| U.S. Census (2018 estimates) | Population density<br>Population<br>African Americans<br>Native Americans<br>Multiracial Americans<br>Hispanic Americans<br>Individuals over 65 years of age<br>Land area | POP_DENSITY<br>TOT_POP<br>BA_MALE, BA_FEMALE<br>NA_MALE, NA_FEMALE<br>TOM_MALE, TOM_FEMALE<br>H_MALE, H_FEMALE<br>ELDERLY_POP<br>Land Area |
| Metric that assesses the vulnerability to COVID-19, taking into account socioeconomic, epidemiological, and healthcare system risk factors | Socioeconomic Status<br>Household Composition and Disability<br>Minority Status and Language<br>Housing Type and Transportation<br>Epidemiological Factors<br>Healthcare System Factors | Socioeconomic Status<br>Household Composition and Disability<br>Minority Status and Language<br>Housing Type and Transportation<br>Epidemiological Factors<br>Healthcare System Factors |
| Features derived from Facebook | Daily mobility relative to average baseline<br>Proportion of users staying in same location | fb_movement_change<br>fb_stationary |
| Epidemiological related Features | Weekly case increase<br><br>Daily tests increase, test positivity<br><br>Projection of case<br>Projection of Rt | confirmed_cases, confirmed_cases_norm, normalized_cases_norm<br>positiveIncrease, positiveIncrease_norm, test_positivity, totalTestResultsIncrease, totalTestResultsIncrease_norm<br>prediction_aligned_int<br>rt_aligned_int |

## Appendix B. SEIR Model

Using human mobility data, a travel network-based SEIR modeling framework (github.com/wpgp/BEARmod, accessed on 10 July 2020) [36] was employed to generate simulated epidemiological data under various outbreak scenarios in Guangzhou and Beijing, where the main parameters were determined in our study (Table A4).

In terms of the epidemiological parameters for the COVID-19 outbreak in Beijing's Xinfadi Market [51], the incubation period was assumed to be a mean of 5.2 days (4.1–7.0) [52]. Due to the illness's high transmissibility during the first five days after onset [53], we calculated the daily contact rate using the basic reproduction number ($R_0$ = 3.32, 1.4–3.9) [54] divided by 5, weighted by the relative level of daily contact based on Baidu movement data (Baidu-based weight). Infectiousness was apparent in an average of two to three days prior to the development of symptoms [55], and the duration from illness onset to isolation of the first case in Beijing was five days. Therefore, the initial lags from infectiousness onset to isolation were set to 7.5 days. The start date of the simulation was set to 3 June 2020, as the first case occurred in Xinfadi Market on that day.

Epidemiological characteristics of SARS-CoV-2 Delta variant infections in Guangdong, China, from May to June 2021, were explored in another study [56]. The mean incubation period was estimated at 5.8 days (95% CI: 5.1–6.5). Owing to 99.8% (93.2–100.0) of transmissions occurring within four days after illness onset, we calculated the daily contact rate using the basic reproduction number ($R_0$ = 4.9, 3.1–6.5) divided by 4, weighted by the daily Baidu-based weight. Patients infected with the Delta variant maintained a high viral load for four days before illness onset, and the number of days from illness onset to isolation of the first case in Guangzhou was two. We then determined the initial number of days from infectiousness onset to isolation to equal 6. The start date of the SEIR model simulation was set to 13 May 2021, considering the first case with symptoms that occurred on 18 May 2021, and the mean incubation period was 5.8 days.

For the outbreaks in Guangzhou and Beijing, we used time lags from the first day of the infectiousness period to the date of isolation as the proxy for the infectious period. The implementation of large-scale nucleic acid testing shortened the infectious period, enabling timely case isolation across the outbreak. The control effects of interventions were expressed by the daily changing contact rate and the shortening of the infectious period. The maximum outbreak duration was assumed to be two months.

The SEIR model was used to simulate the cumulative number of cases in a subdistrict, and the mean of many simulations (e.g., 500) was used to estimate the regional SEIR-based infection risk. The affected subdistricts were estimated by rounding up the infection risk during an outbreak. We employed the constructed SEIR model to estimate affected subdistricts throughout actual COVID-19 outbreaks in Beijing and Guangzhou. Moreover, we used SEIR to generate simulated transmission data under various outbreak scenarios in the two cities.

**Table A4.** Parameters in the epidemiological model (SEIR). To identify subdistricts affected by the actual COVID-19 outbreak in Beijing and Guangzhou, the SEIR model was used to generate simulated COVID-19 outbreaks in the two cities as the sample data.

| Parameter | Beijing | Guangzhou |
|---|---|---|
| Basic reproduction number | 3.32 (95% CI: 1.4–3.9) | 4.9 (3.1–6.5) |
| Incubation period | 5.2 days (4.1–7.0) | 5.8 days (5.1–6.5) |
| Days from illness onset to isolation | 5 | 4 |
| Infectious period | 7.5 (Initial) | 6 (Initial) |
| | Shortened with the implementation of large-scale nucleic acid testing | |
| Start date of the SEIR model simulation | 3 June 2020 | 18 May 2021 |
| Intervention intensity | Relative level of daily contact based on Baidu movement data | |

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
