# Peer review of "Data-Driven Models Informed by Spatiotemporal Mobility Patterns for Understanding Infectious Disease Dynamics"

_ijgi, doi:10.3390/ijgi12070266_

Round 1

Reviewer 1 Report

I really liked this paper.  I really liked the Mobility -informed Risk Index (figure 1). It might be good to add a reference to the first map of infection spread (A2).  In Figure 7, the authors must figure out how to better show it.  The print is too small.

Reviewer 2 Report

Thank you for the opportunity to review this paper. This is very important work that makes great contributions to public health in understanding and predicting infectious disease transmission. I have one minor comment noted in the attached document related to the presentation of some results. Well done.

Reviewer 3 Report

The manuscript " Data-Driven Models Informed by Spatiotemporal Mobility Pat-2 terns for Understanding Infectious Disease Dynamics" uses data driven models to propose mobility-informed risk indices to detect outbreaks caused by emerging acute diseases and predict prevalence trends of diseases. The contribution of this manuscript is of great significance to the current literature where there are not many similar studies on the subject area. The manuscript also highlights the importance of machine learning models in disease early detection, control and surveillance.

 The study also developed two mobility-informed risk indices to describe the risk of infectious disease transmission in space and times which is of great importance to the field of public health.

 The structure is well presented for good readability. The authors do a good job using the right methodological approach to address their question. The results are correctly interpreted and include some practical recommendations for policymakers.

 Below are some minor comments:

 Line 196: Are the township-level divisions (sub-divisions) the smallest possible unit available? If yes, please make it clear in the manuscript. If not, what is the smallest unit available and why did the authors not use that.

Line 231: What is R0? reproductive number? Please make sure to explain what the Ro means for the reader to understand.

Line 264-266: please cite similar studies/sources that calculated sensitivity and specificity similar to how it is presented in the manuscript.

Reviewer 4 Report

Dear,

The paper employed an index considering mobility-informed risk as alternative to data-driven approaches to model infectious diseases dynamics, and also included two other approaches for space and time.

However, on Abstract the Methodology used wasn’t clear and the data doesn’t describe.77%(95% CI)What means CI?

On the abstract the results presented are clear, but the consequences and impacts that such results bring require some explanation.

The authors make clear the proposal in the Introduction section, through the explanation about the mobility risk analysis. But, in my opinion one of the most steps on the papers is to review the past papers, highlighting why the proposal brings the a something innovative related theme. The authors need to pay attention that outbreak Covid-19 allowed to several publications and the risk of contamination by the spread belong to the trend topics. Finally, the review of the past papers is required to improve the quality of paper.

The conclusion hasn't the alignment with the objective proposed initially. For instance, which actions or public policies are required to avoid the increase of the transmission considering your model? Is it the same if we use the SEIR?

Reviewer 5 Report

I found the article interesting, but I had doubts about whether it wouldn't be more complicated to obtain the data to carry out the analyzes in other countries or regions of the world.
